# Cystatin M/E Ameliorates Multiple Myeloma-Induced Hyper Osteolytic Bone Resorption

**DOI:** 10.3390/cancers17050833

**Published:** 2025-02-27

**Authors:** Dongzheng Gai, Perry C. Caviness, Oxana P. Lazarenko, Jennifer F. Chen, Christopher E. Randolph, Zijun Zhang, Yan Cheng, Fumou Sun, Hongwei Xu, Michael L. Blackburn, Guido Tricot, John D. Shaughnessy, Jin-Ran Chen, Fenghuang Zhan

**Affiliations:** 1Myeloma Center, Winthrop P. Rockefeller Cancer Institute, Department of Internal Medicine, University of Arkansas for Medical Sciences, Little Rock, AR 72205, USA; gedongzheng@hotmail.com (D.G.); zzhang4@uams.edu (Z.Z.); ycheng@uams.edu (Y.C.); fsun@uams.edu (F.S.); hxu@uams.edu (H.X.); gjtricot@uams.edu (G.T.); jdshaughnessy@uams.edu (J.D.S.J.); 2Arkansas Children’s Nutrition Center, Little Rock, AR 72205, USA; pccaviness@uams.edu (P.C.C.); oplazarenko@uams.edu (O.P.L.); blackburnmichaell@uams.edu (M.L.B.); 3Department of Pediatrics, University of Arkansas for Medical Sciences, Little Rock, AR 72205, USA; 4Undergraduate Pre-Medical Program, University of Arkansas at Fayetteville, Fayetteville, AR 72701, USA; jfchen@uark.edu; 5Center for Translational Pediatric Research, Arkansas Children’s Research Institute, Little Rock, AR 72202, USA; cerandolph@uams.edu

**Keywords:** bone biology, bone disease, multiple myeloma, hematology, osteoclast/osteoblast biology

## Abstract

The osteolytic lesion is a hallmark of multiple myeloma (MM), caused by increased osteoclast differentiation in the bone marrow. Cystatin E/M (CST6), a cysteine protease inhibitor, was previously discovered by our group to inhibit osteoclast differentiation, making it a potential treatment for MM-induced bone disease. In this study, our group successfully demonstrated that treatment with recombinant CST6 (rmCST6) and zoledronic acid (ZA) in MM tumor-bearing mice and ovariectomy (OVX)-induced hyper-bone resorptive mice significantly reduced bone loss and decreased OVX-induced inflammatory cytokine expression. Additionally, rmCST6 treatment improved bone volume in mice and reduced the proportion of pre-osteoclasts in the bone marrow compared to control and ZA-treated mice. We also found that rmCST6 treatment increased the expression of estrogen receptor alpha (ERα) and the intracellular estrogen concentration in pre-osteoclasts, thereby inhibiting osteoclast maturation.

## 1. Introduction

Multiple myeloma (MM) is a malignancy of terminally differentiated B-cells that is localized primarily in the bone marrow (BM) but also can be present in peripheral blood and tissue/organs. MM cells expand in the BM, produce extra and abnormal proteins, and may crowd out healthy BM cells, suppressing BM function [1]. MM symptoms include hypercalcemia, anemia, renal insufficiency, and osteolysis. Osteolysis, a hallmark of MM, is the cause of severe complications seen in nearly 80% of MM cases and is the result of interactions between MM cells and the BM microenvironment, leading to increased osteoclast differentiation and suppressed osteoblast differentiation, resulting in increased bone resorption and the presence of osteolytic bone lesions [2,3]. These lesions frequently cause bone pain and lead to complications, including pathological fractures, vertebral collapse, spinal cord compression, hypercalcemia, and generalized osteoporosis, compromising MM patients quality of life, impairing survival odds, and increasing treatment costs for MM patients [4].

Bisphosphonates (BPs) are extensively utilized in clinical practice for the treatment of diseases associated with high bone resorption, such as osteoporosis, Paget’s disease, and cancer-induced bone disease [5,6,7]. BPs are pyrophosphate analogs that bind to exposed bone areas of hydroxyapatite crystals [8]. During bone remodeling, they are absorbed by osteoclasts, and through inhibition of intracellular farnesyl pyrophosphate synthase as well as the suppression of GTPase prenylation and interference with downstream pathways, BPs suppress the formation of osteoclasts from precursors and induce apoptosis of mature osteoclasts [9,10].

Based on their chemical structure, BPs are divided into two main groups: nitrogen and non-nitrogen-containing [5]. Among BPs, zoledronic acid (ZA) is the most extensively used in treating cancer-induced bone disease [11]. In subsets of MM patients, clinical trials showed that ZA combined with other novel anti-MM agents reduced skeletal-related events (SREs), prolonged the period between remission and recurrence, and improved overall survival [12,13]. However, although ZA and other anti-resorption drugs are effective at inhibiting osteoclastic bone resorption, the inability of these compounds to repair existing osteolytic lesions and the potential adverse side effects associated with long-term use of such anti-bone resorption drugs, including renal impairment and osteonecrosis of the jaw (ONJ), necessitate the development of novel agents [3,14,15].

Other anti-resorption drugs, like the RANKL monoclonal antibody Denosumab, were approved by the FDA to treat MM bone disease; however, the same adverse effects seen with ZA are also found in treated patients with Denosumab [15]. Recently, we combined PET-CT scanning with global gene expression profiling of BM CD138-selected plasma cells (PC) from 512 newly diagnosed MM patients to show that the absence of osteolytic lesions is linked to elevated expression of cystatin M/E (CST6), a cysteine protease inhibitor, secreted by MM cells. Recombinant CST6 protein inhibits the activity of the osteoclast-specific protease cathepsin K, blocks osteoclast differentiation and function, and inhibits bone destruction in ex vivo and in vivo myeloma models. Suppression of cathepsin L blocks cleavage of p100 to p52 as well as degradation of TRAF3, suppressing the alternative NF-κB pathway [16]. Furthermore, Li et al. reported that CST6 and CST6 peptides (containing the conserved QLVAG residues) inhibit breast cancer bone metastasis by suppressing cathepsin B activity [17]. However, the effectiveness of CST6 in inhibiting MM-induced hyper osteolytic bone resorption relative to bisphosphonates remains unknown. In the absence of such knowledge, a scientific framework upon which to base the identification and development of novel targets for ameliorating MM-induced bone loss will remain unlikely.

In our current study, we utilized both MM- and ovariectomy (OVX)-induced osteolytic and hyper bone resorptive mouse models to compare the effects of ZA and recombinant mouse CST6 (rmCst6) on bone resorption. Single-cell RNA-seq was used to show BM cell populations in murine MM mice following treatment with either rmCst6 or ZA. Previously, it was shown that in breast cancer, loss of CST6 led to a subsequent loss of estrogen receptor alpha (*ERα*) [18]. Estrogen is an important regulator of bone turnover, decreasing the rate of bone resorption and increasing the rate of bone formation, and its actions on bone cells are carried out through interactions with estrogen receptors such as ERα and ERβ. We thus decided to investigate the CST6 effect on intracellular estrogen concentration in osteoclast precursors on protecting against bone deterioration in both MM osteolytic bone disease mouse models and OVX mouse models. The effect of CST6 on estrogen transport and estrogen-related genes in bone tissue, as well as osteoclast precursors, was also investigated. It is our hope that this research will assist in the development of novel bone anti-resorption drugs for the treatment of MM osteolytic lesions as well as for other bone resorption disorders such as osteoporosis.

## 2. Results

### 2.1. Evaluate rmCst6 Protein and ZA Inhibition in MM Cell-Induced Bone Resorption In Vivo

The 5TGM1-KaLwRij murine MM model was utilized to compare the effect of CST6 protein and ZA in vivo on treating MM-induced bone disease. One million 5TGM1 cells were inoculated into C57BL/KaLwRij mice via the tail vein, and mice were treated with purified rmCst6, ZA, or PBS (Figure 1A). Intravenous (i.v.) injection of purified rmCst6 protein (200 μg/kg, twice per week) and subcutaneous (s.c.) injection of ZA (100 μg/kg, twice per week) improved bone volume and density compared with PBS injection, while also significantly decreasing the number of osteolytic lesions in MM-bearing mice (Figure 1B,C). µCT reconstruction of mouse tibia showed that rmCst6 protein and ZA significantly increased trabecular bone volume over total volume (BV/TV), trabecular number (Tb.N), and bone mineral density (BMD). However, rmCst6 or ZA treatment of MM mouse models had no significant effect on trabecular thickness (Tb.Th) or trabecular separation (Tb.Sp) (Figure 1B,C). Additional data for MM mouse model µCT results are listed in Table 1. Histomorphometric analyses demonstrated that rmCst6 and ZA administration significantly reduced osteoclast (OC) numbers as well as the proportion of bone surface occupied by osteoclasts in MM-bearing mice (Figure 1D,E). ZA treatment caused the appearance of bone to be somewhat osteopetrotic when compared with rmCst6 (Figure 1B). To determine whether CST6 or ZA influenced MM tumor burden, flow cytometry was performed to detect the bone marrow GFP+5TGM1 cells in the tibiae at the time of sacrifice. There was no difference between MM-bearing mice and MM mice treated with rmCst6 or ZA (Figure 1F,G). Furthermore, ELISA measurements of tumor-specific M protein, IgG2b, in serum from MM-bearing mice with or without rmCst6 and ZA therapy after 25 days revealed no difference between the control and either treatment group (Figure 1H). Finally, ELISA analyses showed that levels of the C-terminal telopeptide of type-I collagen (CTX-1), which is a biomarker of the rate of bone turnover and osteoclast activity, were significantly reduced in mice treated with ZA and rmCst6 (Figure 1I).

### 2.2. Evaluate rmCst6 Protein and ZA Effects in an Ovariectomized (OVX) Mouse Model

To confirm the ability of CST6 to inhibit osteoclastic bone resorption, we next determined if CST6 could inhibit bone loss in a murine model of estrogen deprivation-induced osteoporosis similar to ZA. Six-month-old C57/BL6 ovariectomized (OVX) mice were treated with PBS, ZA, or rmCst6 for 6 weeks (Figure 2A). After 6 weeks, mice were sacrificed, and tibias were analyzed using μCT and histology. Compared with the sham group, tibias from the OVX mice exhibited significant bone loss, and treatment with either rmCst6 or ZA appeared to improve bone quality back to sham surgery levels or better (Figure 2B,C). Quantitative analysis confirmed that bone parameters, including BV/TV, Tb.N, and BMD, improved in the OVX + rmCst6 mice compared with OVX mice. BV/TV, Tb.N, Tb.Sp, and BMD bone parameters improved in OVX+ZA mice compared with just OVX mice. Tb.Th was not improved in OVX mice after treatment with ZA or rmCst6 (Figure 2C). Similar to the MM mouse model, μCT demonstrated a thick band of calcified trabeculae with distorted architecture was somewhat osteopetrotic under the growth plate in OVX mice treated after ZA (Figure 2B). Data for OVX mouse model µCT results are listed in Table 2. Histomorphometric analyses demonstrated that rmCst6 administration significantly reduced the number of TRAPase-positive cells in OVX mice. However, after 6 weeks of ZA treatment, quantitative statistical histomorphometry showed a significant increase in TRAPase-positive cells in the trabecular bone region, especially on the surface of calcified cartilage that fills the tibia metaphysis (Figure 2D,E), indicating ZA may have unwanted effects on cartilage. The number of adipocyte-like cells in the tibia metaphysis was also measured; there was an initial rise after OVX that was reduced to sham levels after treatment with rmCst6 and to below sham levels after treatment with ZA (Figure 2F). Finally, ELISA analysis showed that CTX-1 and P1NP levels were significantly reduced in mice treated with ZA and rmCst6 protein following OVX compared with OVX mice alone (Figure 2G,H).

### 2.3. Cell Composition of MM Mouse Bone Marrow Is Altered by rmCst6 and ZA Treatment

Single-cell RNA sequencing (scRNA-seq) was used to examine the effect of ZA and rmCst6 on the bone marrow (BM) cell composition in MM mouse models. Based on expression, key gene cells were sorted into specific categories. The uniform Manifold Approximation and Projection (UMAP) plot of BM mononuclear cells indicated that MM cells induced an increase in the percentage of monocyte progenitors and a decrease in B cell percentage in BM (Figure 3A,B). When compared with other groups, only rmCst6-treated mice showed a decrease in BM macrophage percentage and an increase in BM monocyte percentage (Figure 3A,B), implicating decreased osteoclast number is due to decreased precursors of osteoclast. ZA-treated mice alone showed an increase in BM mature neutrophil percentage (Figure 3A).

Since it is known that macrophages are osteoclast precursors, the change in the percentage of previously identified macrophage subtypes was also investigated using UMAP plots of BM macrophages following MM mouse treatment with either PBS, rmCst6, or ZA [16]. rmCst6 and ZA treatment was found to decrease percentage of M0 and M4 macrophages, identified as early precursors of osteoclasts and tumor-associated macrophages with high expression of osteoclast differentiation regulators (*Jun* and *c-Fos*) [16,19]. rmCst6 and ZA also decreased the percentage of M5 macrophages, while ZA alone decreased the percentage of M3 macrophages (Appendix A). rmCst6 treatment was unexpectedly shown to increase the percentage of M7 macrophages, while ZA treatment also increased the percentage of M2 macrophages. M3 macrophages produce inflammatory cytokines and are thought to have tumor-suppressing abilities [20]. M1, M2, M5, and M7 macrophages were classified as being involved in neurological disorders and viral infections but not osteoclastogenesis [16].

### 2.4. Effects of rmCST6 and ZA on the Viability and Differentiation of Osteoclasts, Osteoblasts, and Chondrocytes

rmCst6 has previously been shown to suppress MM-induced osteolytic bone disease by interfering with osteoclast differentiation and function but not viability [16]. On the other hand, ZA was shown to suppress skeletal-related events in MM-induced osteolytic bone disease through inactivation and apoptosis of osteoclasts [1]. To compare the effects of rmCst6 and ZA on suppressing osteoclastogenesis, mouse BM monocytes were induced to differentiate into osteoclasts by the addition of M-CSF (10 ng/mL) and RANKL (10 ng/mL) in the presence or absence of rmCst6 or ZA. TRAPase staining showed that rmCst6 and ZA significantly suppressed the formation of TRAPase-positive multinuclear osteoclasts in a dose-dependent manner (Figure 4A). However, different from rmCst6, ZA appears to be more effective at promoting cell death, as no cells (osteoclasts or precursors) are present at 5 µM ZA (Figure 4A,B). Perhaps the ZA-specific effects on osteoclastic cells might explain ZA-related osteonecrosis.

While ZA was shown to effectively suppress osteoclastogenesis through cell death, there is controversial evidence suggesting ZA can impact osteoblast function [5,10]. To evaluate the effects of rmCst6 and ZA on osteoblast cell viability and differentiation, mouse osteoblast progenitor MC3T3-E1 cells were exposed to different doses of ZA and 200 ng/mL rmCst6, the most effective dose for preventing osteoclast differentiation. Alkaline phosphatase (ALP), a relatively early marker of osteoblast differentiation, staining was performed to assess the effects of ZA and CST6 on the MC3T3-E1 differentiation. Following 14 days of treatment, ALP staining showed that ZA, but not CST6, significantly suppressed the osteoblast differentiation (Figure 4C). The mineralization capacity of cultured osteoblasts treated with ZA and rmCst6 for 21 days were also evaluated using Alizarin Red assay. Again, ZA, but not rmCst6, significantly inhibited the formation of mineralized nodules in a dose-dependent manner (Figure 4C). Because of the effect of ZA on pre-osteoblast differentiation and mineralization, cell viability of MC3T3-E1 cells treated with increasing concentrations of ZA was analyzed. CCK-8 cell viability assay showed that ZA had a significant dose-dependent decrease in cell viability starting at 1 µM (Figure 4D). These data suggest that compared with rmCst6, ZA decreased the osteoblast activity and function.

In addition to osteoblastogenesis, endochondrogenesis is also critical for bone formation. To assess the effect of rmCst6 or ZA on chondrogenesis, teratocarcinoma stem cell line ATDC5 was utilized to determine the effect of both treatments on pre-chondrocyte differentiation, mineralization, and cell viability [21]. Following 14 days of treatment in a chondrocyte differentiation medium, Alcian blue staining demonstrated that the glycosaminoglycan (GAG)-rich extracellular matrix (ECM) found during chondrogenesis was suppressed by ZA treatment but not rmCst6 (Figure 4E). ATDC5 cells can mineralize the surrounding ECM to produce mineral nodules. As such, alizarin red staining was performed, and it was shown that ZA treatment and not rmCst6 inhibited ATDC5 mineralization (Figure 4E). Finally, the CCK-8 cell viability assay showed that similar to osteoblast precursors starting at 2 μM ZA, ATDC5 cell viability was significantly decreased in a dose-dependent manner (Figure 4F).

### 2.5. rmCst6 and ZA Bring OVX-Induced Inflammatory Cytokine Levels Back to Control

Inflammation is a key factor in osteoclastogenesis; as such, the anti-inflammatory effects (and thus the anti-bone resorptive effects) of the treatments used in this study (rmCst6 and ZA) as well as the proinflammatory effects of OVX were measured using an inflammatory cytokine array The 29 targets with the largest initial change in membrane intensity from the sham surgery group to the OVX-PBS group are shown (Figure 5). The anti-inflammatory effects of rmCst6 and ZA appear to be equivalent. Data and statistics for nine targets with the largest initial change in membrane intensity following OVX are listed in Table 3.

### 2.6. ZA and rmCst6 Treatment Upregulate Different Genomic Pathways in Bone of OVX Mice

For the OVX model, it was shown that both rmCst6 and ZA ameliorated OVX-induced inflammation (and thus potentially bone resorption) back to sham surgery levels. To determine the mechanisms by which rmCst6 and ZA suppress bone resorption, sequencing analysis of RNA isolated from the tibia of OVX mice treated with either rmCst6 or ZA was performed (Figure 6A). Between the OVX-PBS and the OVX-rmCst6 treatment groups, there were 175 differentially expressed genes unique to the rmCst6 treatment; between the OVX-PBS and the OVX-ZA treatment groups, there were 12 differentially expressed genes unique to the ZA treatment, and there are three differentially expressed genes shared between both OVX-PBS vs. OVX-rmCst6 and OVX-PBS vs. OVX-ZA (Figure 6B). The 10 genes with the largest fold compared with OVX-PBS are listed for both the rmCst6 and ZA treatment (Figure 6C,D). Of particular interest, for rmCst6 treatment, the solute carrier organic anion transporter family member 1a4 (*Slco1a4*), an organic anion transporter that also facilitates estrogen influx, had the largest fold increase (log2 fold = 3.60), while for ZA treatment, *MMP9*, which plays a role in apoptotic pathways, had the greatest fold increase (log2 fold = 3.33) [22,23]. Since ZA is known for its apoptotic effects on bone cells, the effect of rmCst6 on *Slco1a4* gene expression was investigated further. Basal *Slco1a4* RNA levels were found for different bone tissues and cells using PCR and agarose gel electrophoresis (Tissue: tibia, vertebrae, isolated bone marrow cells, and empty femur; Cell types: Raw 264.7, ST2, OB6, ATDC5, and MLOY4) (Figure 6E,F). Real-time PCR for RNA isolated from OVX mice L3–L5 vertebrae matched RNA-seq results; mRNA levels were decreased following OVX, and only rmCst6 treatment brought *Slco1a4* mRNA back to sham surgery levels (Figure 6G). To determine cell types where rmCst6 treatment increases Slco1a4 gene expression, real-time PCR was performed for macrophage Raw 264.7 cells (±20 ng/mL RANKL) and stromal ST2 cells (±1 mM A2P). For Raw 264.7 cells, only rmCst6 (+20 ng/mL RANKL) displayed a dose-dependent increase in *Slco1a4* mRNA levels up to 100 ng/mL rmCst6 (Figure 6H). rmCst6 treatment also increased Slco1a4 mRNA levels in osteoblast precursor ST2 cells (±1 mM A2P); however, based on one-way ANOVA, changes were not considered significant (Figure 6I,J). Normalized log(2) gene expression for RNA-seq data is listed in Appendix A.

### 2.7. rmCst6 Treatment Increases Intracellular Estrogen Concentration of Osteoclast Precursors

Due to structural and sequence similarities in the solute carrier organic anion transporter family, *Slco1a4* is predicted to play a role in Na^+^ independent transport of estrogen and its derivatives across the plasma membrane into the cell [24,25,26]. Previously, it was found that loss of CST6 in breast cancer led to a loss of ERα expression [18]. As such, we investigated the effect of rmCst6 treatment on estrogen transport and its downstream effects. Protein expression of ERα in L3–L5 was investigated for mice subject to sham surgery or OVX and treated with ZA or rmCst6. When compared with sham surgery mice, ERα protein levels are decreased for OVX mice and brought back to near sham levels only by treatment with rmCst6 but not ZA (Figure 7A). ERα mRNA and protein levels were analyzed at the cellular level using Raw 264.7 cells (±20 ng/mL RANKL). For Raw 264.7 cells incubated in the presence and absence of RANKL, treatment with rmCst6 significantly increased ERα mRNA levels compared with control. For Raw 264.7 cells incubated in the absence of RANKL, 1 µM ZA also increased ERα mRNA levels. ERα protein levels were also increased for Raw 264.7 cells (±20 ng/mL RANKL) (Figure 7B,C). Protein band intensity data quantified by imageJ and normalized to β-actin intensity are listed in Table 4 and Table 5.

Total intracellular estrogen (estrone, E1; 17-β-estradiol, E2; or estriol, E3) concentration was evaluated in non-adherent mouse bone marrow cells (pre-osteoclasts) isolated from the femur following 72 h treatment with either ZA or 100 ng/mL rmCST6. Intracellular total estrogen concentration was significantly increased following treatment with 100 ng/mL rmCst6. Treatment with ZA did not significantly alter intracellular estrogen concentration compared with control (Figure 7D). Immunohistochemistry was used to investigate alterations in estrogen responsive genes in Raw 264.7 cells treated with 100 ng/mL rmCST6. At 24 h incubation, ERα expression was increased following rmCst6 treatment. PPARγ expression, which is suppressed by estrogen, was lowered following rmCst6 treatment (Figure 7E). Raw 264.7 cells were incubated for 7 days with 20 ng/mL RANKL to determine how rmCst6 affects the expression of estrogen-responsive genes in macrophage-like cells differentiating into osteoclasts. Similar to the 24 h incubation period, treatment with 100 ng/mL rmCst6 caused an increase in the expression of ERα and a decrease in the expression of PPARγ (Figure 7F). We validated the remarkable influx of estrogen into Raw 264.7 cells after treatment with 100 ng/mL rmCst6 for 24 h (Figure 7G).

## 3. Discussion

In this study, we show that rmCst6 treatment recovers bone mass in both MM and OVX mouse models without the increased trabecular thickness comparable to the commonly prescribed bisphosphonate ZA. rmCst6 was shown to be as effective as ZA at suppressing osteoclast cell proliferation and viability without cytotoxic effects toward osteoblast and chondrocyte cell lines. When compared with ZA, rmCst6 was shown to significantly increase intracellular estrogen concentration in mouse bone marrow cells. In both male and female bone, estrogen plays an important role in regulating bone turnover, impacting both osteoclast and osteoblast function through a variety of different pathways [27,28,29,30]. As such, biomolecules with the ability to regulate estrogen influx in bone cells would be highly prized as novel agents for combating diseases associated with increased bone resorption.

Based on UMAP results for MM mice, estrogen may attenuate monocyte polarization into macrophage cell type through suppressing expression of key inflammatory factors, explaining the rmCst6-induced increase in monocyte cells and decrease in macrophages [16,31,32,33]. A depletion in macrophage cell percentage would, in turn, lead to a depletion in osteoclasts and, thus, suppression of bone resorption. Both rmCst6 and ZA treatments were also shown to affect the different macrophage subtypes present in the bone marrow. M0, M4, and M5 macrophage subtypes were significantly decreased in both rmCst6 and ZA-treated MM mouse models. M0 macrophages are undifferentiated macrophages with the potential to polarize into different macrophage subtypes, including osteoclasts [19]. The specific role of M4 macrophages is currently unknown. However, this subtype has previously been shown to have increased expression of osteoclast differentiation genes [16]. As with the decrease in total BM macrophage cell percentage, the decrease in M0 and M4 macrophage subtypes following rmCst6 treatment can potentially be explained as estrogen preventing monocyte polarization to macrophage cell types and macrophage polarization into osteoclasts [16,31,32,33,34]. For ZA treatment, a decrease in M0 and M4 macrophages may be from the known apoptotic effects of ZA [22]. In the context of MM osteolysis, as well as other bone resorption disorders, suppression of macrophage cell percentage, as well as a decrease in M0 and M4 macrophage subtypes, would explain the bone protective effects of both rmCst6 and ZA.

Currently, the decrease in M5 and M3 macrophages following ZA treatment can be explained in the context of MM. For ZA, apoptosis may again explain the decrease in M5 and M3 macrophages. Potential apoptosis of anti-tumor acting M3 macrophages hints that ZA alone may not be most effective at treating MM osteolytic bone disease [20,22]. For rmCst6, treatment suppression of M5 macrophages and increased expression of M7 macrophages are more difficult to explain. As M5 and M7 macrophages are thought to play a role in the immune system and there are as precursor of mature osteoclasts, Cst6 may impact M5 and M7 macrophage levels through inhibition of key cysteine proteases involved in immune system regulation [35]. However, it will be needed to clarify the precise mechanism explaining the rmCst6-induced increase in M5 and M7 macrophages. M2 macrophage involvement in oxidative stress repair pathways may explain their increase following ZA treatment due to increased ROS [22,36].

To further investigate the ability of rmCst6 as an inhibitor of bone resorption, rmCst6’s impact on osteoclastogensis on mouse primary non-adherent BM cells was compared with ZA. rmCst6 was found to have a dose-dependent effect on osteoclast number, while ZA, even at low concentrations, significantly depleted the number of osteoclast and precursor cells. In addition to osteoclasts, ZA was also found to have significant suppressive effects on cell vitality of osteoblast (MC3T3-E1) and chondrocyte (ATDC5) precursors, hinting towards toxicity from long-term ZA use [37,38]. In contrast to ZA, rmCst6 at the highest concentration available (200 ng/mL) had no cytotoxic effects on osteoblast and chondrocyte cell vitality, further hinting towards its potential as an alternative to ZA. However, the efficacy, bioavailability, and long-term safety of rmCst6 compared with ZA will need to be addressed in future studies.

In the OVX model, both rmCst6 and ZA were shown to decrease OVX-induced increases in inflammatory cytokines back to sham surgery levels. While an inflammatory cytokine array was performed for only the OVX model, in the context of sex steroid deficiency as well as MM-induced bone loss, suppression of inflammatory factors would suppress osteoclastogenesis and, thus, bone resorption. Sex steroid deficiency is known to cause an increase in adipocyte-like cells present in the bone marrow, explaining the increase in adipocyte-like cells from OVX mouse model histology results. This increase in adipose tissue would, in turn, cause an increase in inflammatory factors and, thus, bone resorption [39]. CST6’s ability to promote estrogen efflux may explain this decrease in adipose tissue [40]. In addition, estrogen has been shown to suppress the expression of inflammatory cytokines such as TNFα, BLC (CXCL13), and SCF [41,42,43]. ZA was also shown to suppress adipocyte-like cell formation and expression of inflammatory factors. In the context of the OVX model, the ZA-induced effect on inflammatory factors may also be through suppression of adipose tissue formation. Currently the mechanism by which ZA reduces adipogenesis is unknown. However, based on previous results for osteoblast and chondrocyte cell lines, high concentrations of ZA may also promote apoptosis of adipocyte-like cells [22]. ZA was also shown to significantly increase the number of TRAP-positive cells in the BM of OVX mice tibia. Increased number of TRAP-positive cells and mRNA levels of *MMP9* may be due to ZA causing an increase in the number of “non-attached” osteoclasts undergoing prolonged apoptosis [22,44,45]. In the future, the impact of rmCst6, as well as ZA, on the expression of inflammatory cytokines in MM mice will need to be studied.

The mechanism by which rmCst6 promotes estrogen influx in macrophage precursors is currently unknown. Based on OVX model RNA-seq data, expression of the organic anion transport protein *Slco1a4* is increased following treatment with rmCst6. In humans, the *Slco1a4* gene analog includes both estrone-3-sulfate and 17β-estradiol-glucuronide as substrates [24,25,26]; they play a role in facilitating estrogen or estrogen receptor signaling. While the OVX mouse model is “sex steroid deficient”, it has been shown that levels of estrone (E1) are higher in post-menopausal women, potentially explaining the bone protective ability of rmCst6 in this model [46]. Based on real-time PCR results from cell lines, upregulation of *Slco1a4* following rmCst6 treatment may partially explain the increase in intracellular estrogen concentration in bone marrow primary cell lines. While Raw 264.7 and ST2 cell lines can differentiate into mature osteoclasts and osteoblasts, respectively, they are not primary precursor cells, and as such, the expression of Slco1a4 will need to be determined in primary osteoclast or osteoblast precursor cells isolated from bone marrow. It will also need to be determined if Slco1a4 is upregulated in MM mice following rmCst6 treatment. To establish *Slco1a4* as a factor in rmCst6-induced estrogen influx, further research is needed.

Cystatins are not thought to make the cell membrane more permeable to biomolecules such as estrogen. To explain the increase in intracellular estrogen levels in osteoclast precursors following rmCst6 treatment, estrogen transport would need to be increased from the upregulated expression of plasma membrane-localized ERα (and potentially Slco1a4). However, a mechanism explaining how rmCst6 increases ERα expression is currently unknown; rmCst6-induced higher Slcola4 activity to facilitate estrogen or estrogen receptor signaling may explain the mechanism. Previously, it was proposed that ERβ may cause an upregulation of cystatins in triple-negative breast cancer [47]. These cystatins may suppress canonical TGFβ signaling, leading to decreased Smad2/3 phosphorylation [48]. Inactivated Smad2/3 decreases transport of Smad4, a known inhibitor of ERα transcriptional activity, into the nucleus [49]. Increased ERα activity may lead to a mechanism whereby ERα regulates its own expression, leading to increased estrogen influx.

We hope that the work performed in this study will assist in the development of novel treatments focused on using recombinant CST6 to ameliorate bone resorption in MM osteolytic bone disease as well as other condition-induced increased bone resorption. Here, we found that increased intracellular estrogen influx in pre-osteoclastic primary BM cells treated with rmCst6 supports a mechanism by which Cst6 mediates increased intracellular uptake of estrogen, leading to decreased bone resorption and increased bone formation [27,28,29,30]. Previously, loss of Cst6 was shown to negatively impact *ERα* gene expression in breast cancer. Yet, the mechanism explaining how Cst6 promotes estrogen influx is currently unknown. It is our belief that this research will lead to further studies focused on understanding the mechanism by which Cst6 is able to promote estrogen influx and suppress bone resorption. Our future efforts to understand the mechanism by which CST6 suppresses bone resorption will be investigated through the use of an ERα conditional knockout mouse model and by investigating TGFβ/Smad signaling pathways that CST6 may potentially regulate.

## 4. Material and Methods

### 4.1. Cell Cultures

Murine osteoclast progenitor macrophage cell line RAW 264.7 cells and bone marrow stromal cell line ST2 cells were commercially obtained (American Type Culture Collection [ATCC], Manassas, VA, USA, http://www.atcc.org). Cells were initially cultured in Dulbecco’s modified Eagle medium (DMEM) supplemented with 10% heat-inactivated fetal bovine serum (FBS) and penicillin (100 µg/mL)/streptomycin (100 µg/mL) (1% P/S). MC3T3-E1 osteoblastic precursor cells were initially cultured in α-MEM, supplemented with 10% FBS, 1% P/S. ATDC5 mouse chondroprogenitor cells were purchased commercially (Sigma Aldrich, St. Louis, MO, USA). ATDC5 cells were initially cultured in DMEM/F12 (containing 2 mM glutamine) (Corning, catalog # 10103CV; Mediatech Inc., Manassas, VA 20109, USA) with 5% FBS and 1% P/S. For all cell types, media was changed every 2–3 days, and cells were split when necessary to avoid over-confluence.

In ex vivo, mouse bone marrow cells from flushed femur were first cultured in α-minimum essential media (αMEM) supplemented with 10% FBS, 1% P/S, and 4 mM glutamine for 24 h. Following this, non-adherent bone marrow cells were cultured onto tissue culture plates for analysis of intracellular estrogen concentration.

### 4.2. 5TGM1/KaLwRij MM Mouse Model

Six to eight-week-old female C57BL/KaLwRij mice (Harlan Mice, Horst, The Netherlands) were injected with 1 × 10^6^ 5TGM1-GFP cells intravenously via the tail vein and randomized into 3 groups (MM mice, *n* = 5 per group). Five days after injection of tumor cells, mice were treated with either PBS (100 µL), rmCst6 (200 μg/kg) via intravenous (i.v.) injection twice a week, or Zoledronic Acid (ZA) (100 μg/kg) via subcutaneous (s.c.) injection twice a week. On day 25 post-tumor cell inoculation, when most mice had started to develop paraplegia, the experiment was terminated, and mice were sacrificed. Blood samples were collected every week. All animal procedures adhered to a protocol approved by the local Institutional Animal Care and Use Committee (IACUC) at the University of Arkansas for Medical Sciences.

### 4.3. Recombinant CST6 (rmCst6) Expression and Purification

Mouse CST6-cDNAs cloned into pcDNA3.1(+)-C-6His was purchased from GenScript (Piscataway, NJ, USA). The constructs were then transfected into HEK293T cells via Lipofectamine2000 (ThermoFisher, Cat#11668500; Life Sciences Solutions, Carlsbad, CA 92008, USA). Conditioned media was collected 48 and 72 h after transfection. The pH of the medium was adjusted to pH7.5–8.0 with 0.05 M NaOH, then loaded into the HisTrap HP column (Cytiva, Cat#17524801; Life Sciences Solutions, Carlsbad, CA 92008, USA) using a peristaltic pump at 4 °C. The His tag proteins were washed with 50 mL of 50 mM Na-Phosphate, 300 mM NaCl, 10% glycerol, 5 mM Imidazole pH 7.5, and eluted with 50 mL 0–100% to 50 mM Na-Phosphate, 300 mM NaCl, 10% glycerol, 300 mM Imidazole pH 7.5 using the NGC column Chromatography System (Bio-Rad, Hercules, CA, USA). After concentration by ultrafiltration, 5 mL samples were loaded onto a Superdex 75 100/300 GL column (Cytiva, Cat#29148721) pre-equilibrated with 50 mM Na-Phosphate pH 7.5, 150 mM NaCl, at a flowrate of 0.75 mL/min. The protein purity was determined by silver stain according to the Pierce Silver Stain Kit protocol (ThermoFisher, Cat#24612; Life Sciences Solutions, Carlsbad, CA 92008, USA). The concentration of the purified protein was determined at 280 nm by nanodrop 2000 (Thermo Scientific). The purified protein was tested for functionality prior to use in in vivo tests.

### 4.4. Ovariectomy (OVX) Mouse Model

Six-month-old female C57BL/6J mice (Jackson Laboratories) were utilized in this study. Mice were anesthetized with chloral hydrate and subjected to ovariectomy or sham operation. Following OVX surgery, mice were randomly divided into 4 groups to receive the following treatments: (1) sham operation + PBS (sham, *n* = 8), (2) OVX + PBS (OVX-PBS, *n* = 8), (3) 100 μg/kg Zoledronic Acid (OVX-ZA, *n* = 9), (4) 50 μg/kg rmCst6 (OVX-CST6, *n* = 10). Two days following surgery, mice were administered drugs for 6 weeks via i.p, i.v, or s.c injection. OVX-ZA mice received injections twice a week, while OVX-CST6 mice received injections every other day. After 6 weeks, mice were sacrificed. Mouse serum, legs, and vertebrae were collected and stored at −80 °C until used. Successful OVX was confirmed by an assessment of uterus weight and weight development during the experiment.

### 4.5. Micro-Computed Tomography (µCT)

µCT analysis of mice tibia was performed as previously described [50]. Tibias from both MM and OVX models were dissected and fixed in 10% neutral-buffered formalin for 2 days. For µCT, region of interest (ROI) was selected to include the entire epiphysis and metaphysis of one end of the tibia as it contains trabecular bone. Scans were acquired at 60 kV and 166 uA; Al 0.5 mm filter; 10 uM pixel size. After scanning, tibia images were reconstructed using the Skyscan NRecon program with a beam hardening correction of 40. Trabecular and cortical bone microarchitecture were analyzed using the Skyscan CT Analyzer program. The following bone parameters were calculated using µCT analysis: bone volume fraction (BV/TV%), trabecular number (Tb N, 1/mm), trabecular thickness (Tb Th, mm), trabecular separation (Tb Sp, mm), and bone mineral density (BMD).

### 4.6. Bone Histomorphometry

Following µCT, tibia from both MM and OVX mouse models were decalcified in 5% EDTA solution (pH 7.0) for 7 days at room temperature and embedded in paraffin. Bone sections (5 μm thickness) were stained for H&E and tartrate-resistant acid phosphatase (TRAPase) staining using a Leukocyte Acid Phosphatase Kit (Sigma-Aldrich, St. Louis, MO, USA). For the MM mouse models, histomorphometric analyses were performed using the OsteoMeasure software (OsteoMetrics, V4.2.1.0) with a Zeiss Axioskop2 microscope (Microscope Central; Willow Grove, PA 19090, USA). For the OVX mouse models, TRAPase-positive cells and adipocyte-like cells were counted using Nikon NIS elements platform and normalized to total area of ROI at 20× magnification (285,474.816 µm^2^). At 10× and 20× magnifications, regions were selected to include areas with minimal damage. In addition to this, regions were also chosen to include all cell and tissue types (adipocyte-like cells, TRAPase-positive cells, and bone matrix). For both MM and OVX mouse models, histomorphometric parameters were averaged for each treatment and graphed using GraphPad prism 9.0.

### 4.7. Evaluation of Intracellular Bone Turnover Markers and Estrogen Levels

The serum levels of CTX-1 and P1NP were examined using a CTX-1 ELISA kit and P1NP ELISA kit (CTX-1: Cat# NBP3-11802, P1NP: Cat# NBP2-76466; Novus Biologicals LLC; Centennial, CO 80112, USA) according to the manufacturer’s instructions for both OVX mice. Intracellular total estrogen concentration for non-adherent BMC treated with rmCst6 (100 ng/mL) or ZA (1 µM) was examined using a mouse estrogen ELISA kit (Mybiosource: MBS766177; MyBioSource, Inc, San Diego, CA 92195, USA). Cells were cultured in αMEM in triplicate with treatments for 72 h at 37 °C. After 72 h, cells were lysed and assayed according to the manufacturer’s instructions. Total intracellular estrogen concentration for non-adherent BMC was measured using triplicate analysis, and the experiment was repeated three times to give *n* = 9 for each treatment (Control, 1 µM ZA, 100 ng/mL rmCst6). Results for estrogen concentration (pg/mL) were normalized by dividing by intracellular protein concentration (µg/mL). To reduce variability between plates, data were normalized for each plate by dividing estrogen concentration (pg/µg) for each sample by the average estrogen concentration of the control treatment for said plate, then multiplying by the average estrogen concentration for all samples across three plates. Following normalization, results were plotted using GraphPad Prism 9.0.

### 4.8. Single Cell RNA Sequencing (scRNA-seq)

scRNA-seq was performed as previously described [16]. Briefly, BM mononuclear cells from 5TGM1/KaLwRij (MM) mice were isolated at 25 days post tumor cell inoculation. Cells were isolated from healthy control mice (*n* = 2), MM mice (*n* = 3), MM mice+rmCst6 (*n* = 3), or MM mice+ZA (*n* = 2) treated mice sorted using fluorescence-activated cell sorting (FACS). Sorted GFP-negative cells with a purity greater than 95% and viability higher than 95% were used for 10× genomics scRNA-seq, resuspended in Mg- and Ca-free PBS (+0.04% BSA), and counted using a light microscope under 10× magnification. Single-cell emulsions were generated (Chromium Controller, Chromium Next GEM Chip G, Chromium NextGEM Single Cell 3’ v3.1 kit; 10X Genomics, Pleasanton, CA, USA). Libraries were assessed for mass concentration and fragment size and validated. Initial sequencing was performed on an Illumina NovaSeq SP 100-cycle flow cell, and data were assessed using the Cell Ranger Count output.

### 4.9. Bioinformatic Analysis of scRNA-seq

Bioinformatics analysis of scRNA-seq data was performed as previously described [16]. Briefly, raw sequencing fastq files were processed using CellRanger software (10× Genomics) version 6 with *Mus musculus* reference genome. The count table was loaded into R through Seurat version 4 package for further analysis. Cells with gene numbers less than 500, greater than 5000, and more than 10% of unique molecular identifiers from mitochondrial genes were discarded from the analysis. Principal component (PCA) was performed on significantly variable genes from remaining cells. Nitration results were used as input for clustering using Louvain algorithm with multilevel refinement and Uniform Manifold Approximation and Projection for Dimension Reduction (UMAP). Gene-specific markers of each cluster were identified using the FindMarkersAll function with MAST method test statistics. Cell clusters and gene markers include mature neutrophils-I (*S100a8*, *Ly6g*), mature neutrophils-II (*Rethlg*, *MMP9*), immature neutrophils-I (*Chil3, Camp*), macrophages (*Adgre1*, *Mafb*), monocyte prog-I (*F13a1*, *Itga1*), monocyte prog-II (*Prtn3*, *Hsp90ab1*), dendritic cells (*Siglech*, *Bst2*, *Tcf4*), lymphomyeloids prog-I (*Irf8*, *Flt3*), myeloid prog (*Mpo*, *Cd63*), immature neutrophils-II (*Cenbf*, *Camp*), B cells (*Ighm*, *Cd79b*), NK/T cells (*Cd3e*, *Gzmb*), immature neutrophils-III (*Fcnb*, *Camp*), monocytes (*ly6c2*, *ccl2*), pre/pro B cells (*Pax-5*, *Vpreb3*), lymphomyeloids prog-II (*Atp2b4*, *Hlf*), and eosinophil/basophil prog (*Gata2*, *Cpa3*).

To visualize genes simultaneously in kernel joint density estimation, the Nebulosa package was used. Based on the kernel joint density of Adgr1 and Fcgr3, we sub-selected cells that have a high value of the kernel joint density for subclustering analysis to study the cellular heterogeneity of macrophage cells. Gene-set enrichment analysis of marker genes was performed on Gene Ontology annotation using piano package.

### 4.10. In Ex Vivo Osteoclast Differentiation

Primary mouse BM cells were flushed out from the femur and tibia of 6–8-week C57BL/6J mice; BM erythrocytes were removed with the ACK Lysing Buffer (KD Medical, catalog # RGF3015) for 3 min. The remaining cells were cultured in α-MEM (Gibco, catalog # 12571048) supplemented with 10% FBS and 10 ng/mL M-CSF (PeproTech, catalog # 315-02) overnight in 10 cm tissue culture dish. A total of 4 × 10^4^ non-adherent cells were seeded into 96-well plate with α-MEM containing 10% FBS and 10 ng/mL M-CSF for 3 days to recruit macrophage, and then osteoclast differentiation was induced by addition of 10 ng/mL RANKL (R&D, catalog # 462-TEC-010/CF) with or without rmCst6 protein or ZA for 3–5 days. Media changes were carried out every 2 days. After 3 days, the cells were evaluated for TRAP staining.

### 4.11. Osteoblast and Chondrocyte Precursor Histological Staining

Osteoblast precursors MC3T3-E1 were seeded at 15,000 cells/cm^2^ in 12-well or 24-well plates (α-MEM, supplemented with 10% FBS, 1% P/S). At 80% confluence, the media was changed to osteogenic induction media (α-MEM, supplemented with 10% FBS, 1% P/S, 50 μg/mL L-Ascorbic Acid (Sigma-Aldrich, catalog # A5960) and 5 mM β-Glycerophosphate (Sigma-Aldrich, catalog # 9422). Media was changed every 2–3 days during the culture period. On day 14, the activity of alkaline phosphatase (ALP) was evaluated using histological staining. Cells were washed with PBS and fixed with 4% paraformaldehyde for 20 min at room temperature and then stained with 1-step NBT-BCIP substrate solution (Thermo Scientific, catalog # 34042) for 20 min. After staining, cells were washed twice with water and observed under a light microscope. On day 21, cells were fixed and stained with 2% Alizarin Red (Sigma-Aldrich, catalog # A5533) at pH 4.2 to evaluate the mineralization using previously described methods [51].

ATDC5 mouse chondroprogenitor cells were purchased commercially (Sigma Aldrich, St. Louis, MO, USA). ATDC5 cells were initially cultured in DMEM/F12 (containing 2 mM glutamine) (Corning) with 5% FBS and 1% penicillin/streptomycin (P/S). For chondrocyte mineralization, cells were initially seeded at 15,000 cells/cm^2^. At 80% confluence, the media was changed to chondrocyte differentiation media comprised of DMEM/F12 (containing 2 mM glutamine) (Corning) with 5% FBS, 1% P/S and 1X insulin-transferrin-selenium (ITS) (ThermoFisher, catalog # 41400045), and cells were treated with either 200 ng/mL rmCst6 or ZA (0.5 µM, 1 µM, 2 µM). The media was changed every 2–3 days during the culture. After 14 days, cells were washed with PBS, fixed with 4% paraformaldehyde (20 min, room temperature), and then stained with Alcian blue staining solution (Sigma Aldrich, catalog # TMS-010-C) for 30 min at room temperature [51]. After staining, cells were washed twice with water and observed under a light microscope. On day 21, ATDC5 cells were fixed and stained with 2% Alizarin Red (Sigma-Aldrich, catalog # A5533) at pH 4.2 to evaluate the mineralization [51].

### 4.12. Cell Counting Kit (CCK)-8 Assay

Two thousand cells (either ATDC5 or MC3T3-E1 cell line) seeded in 96-well plate in 100 μL culture medium were allowed to adhere overnight and then were treated with different doses of Zoledronic Acid (0–20 μM) for 7 days in a humidified atmosphere of 5% CO_2_ at 37 °C. Following treatment, 10 μL CCK-8 solution (Apexbio Technology, catalog # K10181) was added to each well, and the plate was incubated for 2 h in the incubator. The absorbance was measured at a wavelength of 450 nm using a microplate reader (BioTek Laboratories LLC, Shoreline, WA 98133, USA). Results were averaged and plotted using GraphPad Prism 9.0.

### 4.13. Cytokine Array

RayBio C-Series Mouse Cytokine antibody array C1000 (RayBiotech, Catalog#: AAM-CYT-1000-2) was used according to the manufacturer’s protocol with protein lysate from ovariectomized mice L3 to L5 vertebrae. Protein was isolated from vertebrae by homogenization with RIPA buffer (Solarbio) followed by sonication. Protein concentrations were found for lysates using BCA assay, and for each treatment, samples were pooled. Final concentration for each pooled sample was 1.94 mg/mL. Samples were added to array membrane in triplicate according to the manufacturer’s protocol. A heat map (GraphPad Prism 9.0) was created for array targets where CST6 or ZA treatments were shown to reduce cytokine levels elevated by OVX to near-sham levels.

### 4.14. RNA Isolation, Real-Time Reverse Transcription-Polymerase Chain Reaction

For measurement of *Slco1a4* mRNA levels, Raw 264.7 cells (20 ng/mL RANKL) or ST2 cells were cultured into 12 well tissue culture plates (2.5 × 10^5^ cells/mL, 1 mL total volume). Cells were treated with either a PBS control solution or increasing concentrations of rmCst6 (50 ng/mL, 100 ng/mL, or 200 ng/mL). Cells were incubated at 37 °C for 72 h. For ST2 cells incubated with ascorbate-2-phosphate (A2P), cells were initially cultured at 2000 cells/well and incubated at 37 °C for 10 days, with the same treatments applied. Media and treatments were replaced every 48 h. For measurement of *ERα* mRNA levels, Raw 264.7 cells were cultured into 12 well tissue culture plates (2.5 × 10^5^ cells/mL, 1 mL total volume), treated with PBS control solution, 1 µM ZA, or 100 ng/mL rmCst6 (±20 ng/mL RANKL), and incubated at 37 °C for 48 h.

For each treatment, triplicate samples were cultured. RNA was isolated using RNeasy plus Mini Kit from Qiagen. RNA concentration and purity (A260/A280) for RNA samples were determined using a Polarstar Omega plate reader. Reverse transcription was carried out using an iScript cDNA Synthesis Kit. Real-time polymerase chain reaction (RT-PCR) experiment was carried out using SYBR Green and the QuantStudio 6 Flex real-time PCR system from Applied Biosystems. Samples were performed in triplicate, and gene expression was normalized to cyclophillin A and averaged for each treatment concentration. ERα, Slco1a4, and cyclophillin A primers for RT-PCR analysis were designed using Primer Express Software 2.0.0.

For analysis of basal *Slco1a4* tissue and cell mRNA levels, RNA was isolated from the following tissues using the trizol method: tibia, L3–L5 vertebrae, bone marrow cells, and empty femur, as well as from the following cell types: Raw 264.7, ST2, OB6, ATDC5, and MLOY4, and purified using RNeasy plus Mini Kit. RNA concentration and purity (A260/A280) for RNA were determined as previously described, and cDNA was synthesized. QuantStudio 6 Flex real-time PCR system was used to determine basal expression in tissues and cell culture. Following real-time PCR, gene expression was then visualized using agarose gel electrophoresis.

For in vivo analysis of Slco1a4 and ERα mRNA levels, total RNA was isolated from mouse L3 to L5 vertebrae for all samples from each treatment group (sham, OVX-PBS, OVX-CST6, OVX-ZA). Tissue was first homogenized using metal beads and a homogenizer (6500 rpm, 20 s, twice). After homogenization, total RNA was isolated using trizol and purified using the RNeasy plus Mini Kit. For real-time PCR, RNA concentration and purity were determined as with previous RNA samples, and cDNA was synthesized using iScript cDNA Synthesis Kit. RT-PCR experiment was carried out using SYBR Green, as described in the above paragraphs, with gene expression normalized to cyclophillin A. For all RT-PCR experiments, results were averaged and graphed using GraphPad Prism 9.0.

### 4.15. Bone Tissue RNA Sequencing Analysis

For RNA sequencing experiments, RNA was isolated from in vivo (tibia; OVX mouse model; sham, OVX-PBS, OVX-CST6, OVX-ZA; three samples per group) samples as described earlier. R12NA integrity was checked using Agilent Technologies 4200 Tapestation with RNA screen tape. Samples with RIN^e^ greater than 8 were used for further analysis. RNA sequencing was performed in the UAMS developmental genomics core. Briefly, sequence-ready libraries were constructed using the Illumina Stranded mRNA Prep Ligation kit following the manufacturer’s protocol and then sequenced on an Illumina NextSeq 500. Briefly, poly-A-containing mRNA was isolated from total RNA using oligo-dt-attached magnetic beads and then converted to cDNA. After fragmentation and end repair, cDNA 3’ ends were adenylated and then ligated with index anchors. The anchor ligated fragments were indexed and amplified, then normalized, pooled and sequenced.

Analysis of RNA sequencing results was performed at the Louisiana Cancer Research Center. Briefly, FASTQ files were uploaded to Partek Flow, and contaminants were removed with Bowtie 2 (v2.2.5). Reads were aligned to STAR v2.7.8a using the mm10 version of the mouse genome. Aligned reads were quantified using Ensembl Transcripts release 102. Features with a maximum reads ≤ 5 were filtered out from the analysis. Normalization was performed by TMM and log2 transformation. Pathway analysis (KEGG) was performed in Partek Flow. Heat maps comparing LS mean for treated samples (OVX-CST6, OVX-ZA) with OVX samples were created using GraphPad Prism 9.0.

### 4.16. Western Blots

To analyze ERα protein expression, total protein from each sample was first isolated from L3 to L5 vertebrae by homogenization with RIPA buffer (Solarbio) followed by sonication. Protein concentrations were found for lysates using BCA assay. For ERα protein expression in Raw 264.7 cells (±20 ng/mL RANKL), cells were cultured onto 6-well tissue culture plates with either control, 1 µM ZA, or 100 ng/mL rmCst6 treatments and incubated at 37 °C for 48 h (2.5 × 10^5^ cells/mL, 2 mL total volume). After 48 h, cells were lysed using RIPA buffer, and protein concentration was found using BCA assay. Both protein tissue and protein cell samples were prepared for SDS PAGE; equal amounts of protein were loaded for each sample and then run on 10% acrylamide/Bis-acrylamide gels (*n* = 6 for bone tissue, *n* = 3 for Raw 264.7 cells). Following membrane transfer, mouse anti h/m/r ERα (MAB57151, R&D Systems) was used as the primary antibody, and HRP-conjugated anti-mouse IgG (R&D Systems, HAF018) was used as the secondary antibody. Bands of interest were visualized and imaged under chemiluminescent detection using the Amersham Imager 600 System. Amido black staining (for tissue lysate samples) and mouse anti β-actin antibody (Sigma, A1978) (for cell lysate samples) were used as loading controls for western blots. Densitometry analysis of western blot bands was performed using ImageJ (Version 1.54g).

### 4.17. Immunohistochemistry

Raw 264.7 cells were grown to 10,000 cells/well in Nunc. Lab Tec Chamber slides (Thermo Fisher, 177399) in DMEM supplemented with 10% FBS and 1% P/S. Cells were treated with either control solution or 100 ng/mL rmCst6 and incubated at 37 °C for either 24 h or 7 days. The 7-day incubation period was in the presence of 20 ng/mL RANKL. Following incubation, immunostaining was performed as follows: Media was aspirated, and cells were rinsed with cold 4% paraformaldehyde in 1X PBS and incubated at room temperature for 1 min. Cells were then fixed with 4% paraformaldehyde in 1X PBS at room temperature for 20 min. Paraformaldehyde was removed, and cells were rinsed with 1X PBS. Cells were then covered with 1X PBS (10–15 min, room temperature). PBS was removed, and 2.5% horse blocking serum was added (20 min, room temperature). After 20 min, blocking serum was removed via aspiration, and cells were incubated with primary antibody (ERα: Millipore, MAB447; PPARγ: abcam, ab191407) diluted 1:50 in 2.5% horse blocking serum containing 1% IGEPAL overnight at 4 °C. Cells were then washed with 1X PBS containing 0.05% IGEPAL (3 min, 3 times at room temperature), and secondary antibody was added (ERα: abcam, ab150105, PPARγ: abcam, ab150067) (1 h, room temperature, protected from light). Cells were washed with 1X PBS containing 0.05% IGEPAL again (3 min, 3 times at room temperature, protected from light). Final cells were covered with DAPI-Fluoromount-G and observed using Nikon Eclipse T/2 epifluorescent microscope.

### 4.18. Statistical Analysis

Statistical analysis was performed with GraphPad Prism 9.0 (GraphPad Software, Inc., San Diego, CA, USA). Numerical variables were expressed as mean ± SD (Standard Deviation); *n* equals the number of samples/group. For all experiments, differences within groups were evaluated using either one-way ANOVA or *t*-test followed by Turkey’s post hoc test. Values were considered statistically significant at *p* < 0.05.

## Figures and Tables

**Figure 1 cancers-17-00833-f001:**
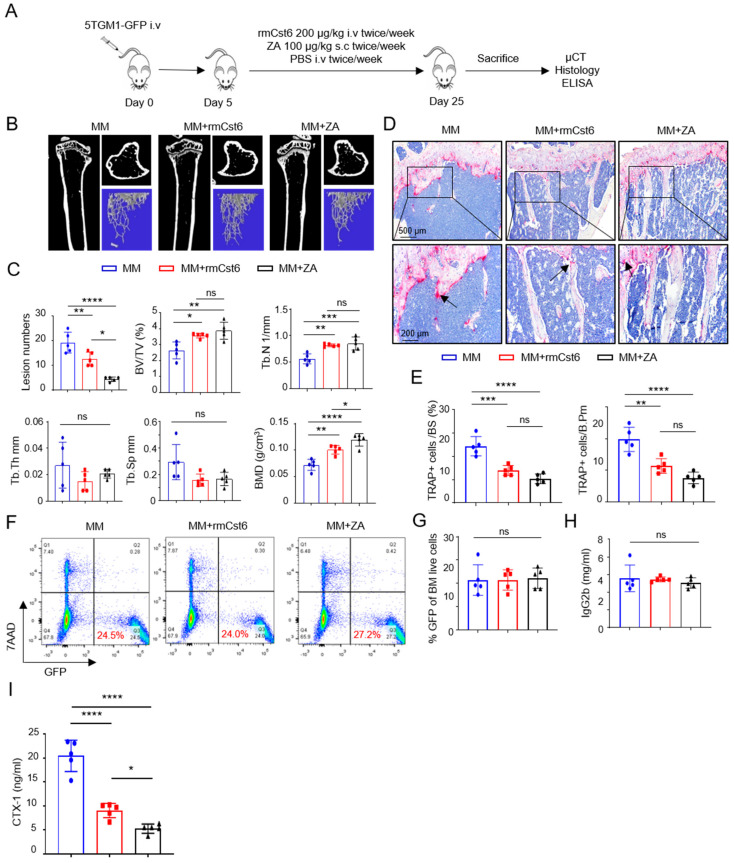
rmCst6 protein and ZA inhibit bone destruction in 5TGM1-C57BL/KaLwRij MM mice. (**A**) A schematic model for the MM mouse study. 5TGM1 murine MM cells were injected into 8-week-old C57BL/KaLwRij female mice via tail vein. Recombinant mouse Cst6 (rmCst6) protein and ZA were administered on day 5 post-tumor inoculation. On day 25, mice were sacrificed, and samples were collected. (**B**) Reconstructed µCT images of tibia sagittal sections taken from MM mice show trabecular architecture change in bone for MM mice and MM mice treated with either rmCst6 or ZA. (**C**) Bar plots present the number of bone lytic lesions on the right medial tibia surface and the trabecular bone parameters: BV/TV, Tb.N, Tb.Th, Tb.Sp, and BMD (*n* = 5). (**D**) Best representative images of TRAPase staining show TRAP-positive cells (indicated with arrows) in tibiae derived from C57BL/KaLwRij mice injected with 5TGM1 MM cells with or without rmCst6 and ZA treatment. (**E**) Bar plots represent histomorphometric analyses of TRAPase-stained number of osteoclast per bone perimeter (N.Oc/B.Pm) and osteoclast surface per bone surface (Oc.S/BS) (*n* = 5). (**F**) Representative flow cytometry plots presented the GFP+5TGM1 cells in BM of control and rmCst6 and ZA-treated mice. (**G**) Bar plots represent the percentage of GFP+5TGM1 cells in BM (*n* = 5). (**H**) Tumor burden was assessed by measuring serum levels of IgG2b (mg/mL) by ELISA (*n* = 5). (**I**) Bar plots show serum levels of the bone resorption marker CTX-1 detected by ELISA (*n* = 5). For all measurements, data are represented as mean ± SD and were analyzed by one-way ANOVA with Tukey’s multiple comparisons. Denoted markings are considered significant * *p* < 0.05, ** *p* < 0.01, *** *p* < 0.001, **** *p* < 0.0001, ns: not significant.

**Figure 2 cancers-17-00833-f002:**
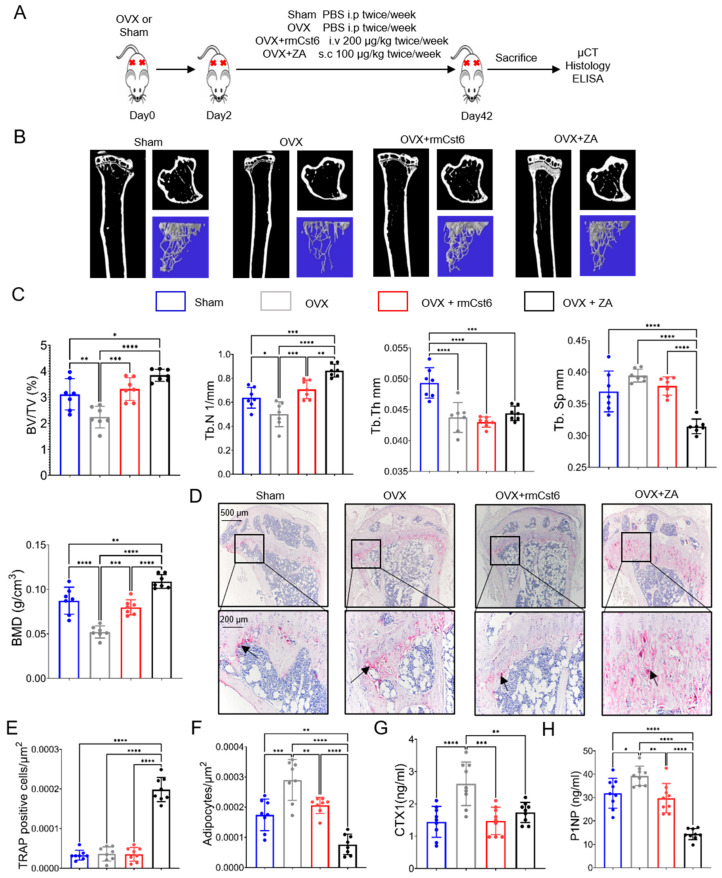
rmCst6 protein and ZA suppress bone resorption in sex steroid deficient ovariectomized (OVX) mouse model. (**A**) A schematic model for the OVX mice study. Six-month-old female mice with OVX were administered rmCst6 protein or ZA for 6 weeks by i.v. or s.c. injection twice a week. After 6 weeks, mice were sacrificed, and samples were collected. The red cross represents the site of s.c. injection. (**B**) Reconstructed µCT images of tibia from OVX mice show trabecular architecture change in bone following sex steroid depletion and treatment with either rmCst6 or ZA. (**C**) Bar plots show the trabecular bone parameters: BV/TV, Tb N, Tb Th, Tb Sp, and BMD (*n* = 7). (**D**) Best representative images of TRAPase staining shows TRAPase-positive cells (indicated with arrows) and adipocyte-like cells present in tibia from Sham, OVX, OVX + ZA, and OVX + rmCst6 mice. (**E**) Bar plots represent the histomorphometric analyses of the number of TRAPase-positive cells in the tibia normalized to total area of ROI at 20×. Number of TRAPase-positive cells/µm^2^ is increased only with ZA treatment (*n* = 8). (**F**) Average number of adipocyte-like cells/µm^2^ shows that adiposity in bone marrow is increased following OVX but brought back to near sham levels by rmCst6 or ZA treatment (*n* = 8). (**G**,**H**) Barn plots show the serum levels of the bone resorption marker CTX-1 and bone formation marker P1NP detected by ELISA from each group are recovered to near or better than sham levels following rmCst6 or ZA treatment (*n* = 9). Data shown as mean ± SD. Statistical analysis was performed using one-way ANOVA. Denoted markings are considered significant * *p* < 0.05, ** *p* < 0.01, *** *p* < 0.001, **** *p* < 0.0001.

**Figure 3 cancers-17-00833-f003:**
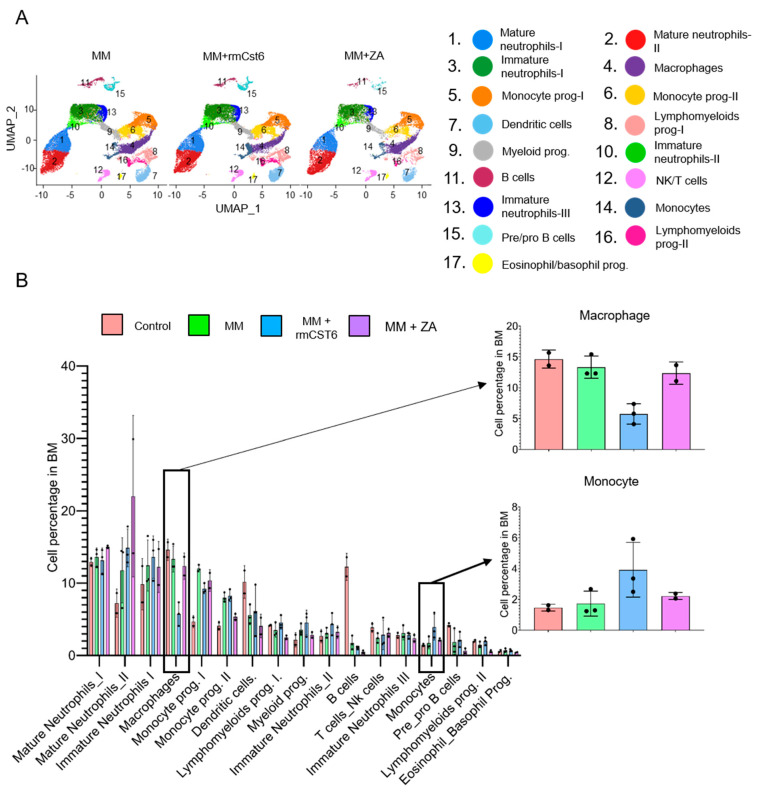
scRNA-seq reveals BM microenvironment alterations after rmCst6 and ZA treatment. (**A**) The Uniform Manifold Approximation and Projection (UMAP) plot of BM mononuclear cells derived from control mice (*n* = 2), MM mice treated with PBS (*n* = 3), rmCst6 protein (*n* = 3), or ZA (*n* = 2). (**B**) Bar views show the proportion of various cell types in BM mononuclear cells of control or MM-bearing mice treated with PBS, rmCst6, or ZA protein.

**Figure 4 cancers-17-00833-f004:**
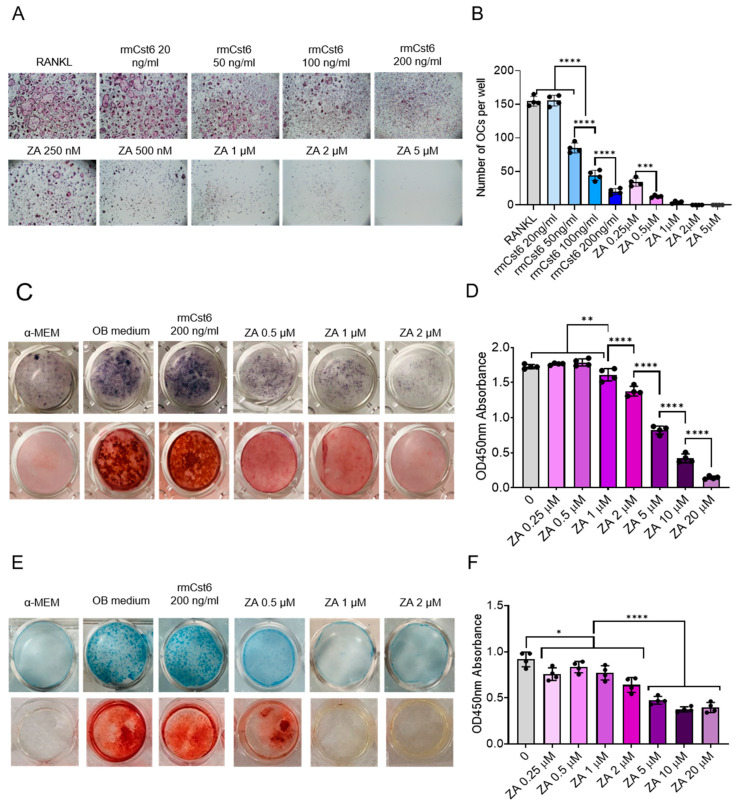
The effects of CST6 and ZA on the viability and differentiation of osteoclast, osteoblast, and chondrocyte. (**A**) Mouse osteoclast precursors were differentiated into osteoclasts by addition of M-CSF and RANKL. Different concentrations of rmCst6 protein and ZA were added into the culture media for 4 days. TRAPase staining shows osteoclasts containing multiple nuclei. (**B**) Bar plots show the quantification of TRAP+ osteoclasts (*n* = 4). (**C**) MC3T3-E1 cells were differentiated to osteoblasts with 50 μg/mL ascorbic acid and β-glycerophosphate. Alkaline phosphatase staining on day 14 (**upper** panel) and Alizarin red staining on day 21 (**lower** panel) showed ZA but not CST6 inhibited osteoblast differentiation and mineralization. (**D**) MC3T3-E1 cells were incubated with different doses of ZA for 7 days, and cell viability was detected by CCK-8 assay (*n* = 4). (**E**) ATDC5 cells were differentiated to chondrocytes with ITS medium for 14 days. Alcian blue staining (**upper** panel) showed the glycosaminoglycan (GAG) deposition was suppressed by ZA; ATDC5 cells were induced with 1X ITS medium plus ZA or CST6 for 21 days. Endochondral ossification was detected by Alizarin red staining (**lower** panel). (**F**) ATDC5 cells were incubated with different doses of ZA for 7 days, and cell viability was detected by CCK-8 assay (*n* = 4). Data shown as mean ± SD. Statistical analysis was performed using one-way ANOVA. Denoted markings are considered significant * *p* < 0.05, ** *p* < 0.01, *** *p* < 0.001, **** *p* < 0.0001.

**Figure 5 cancers-17-00833-f005:**
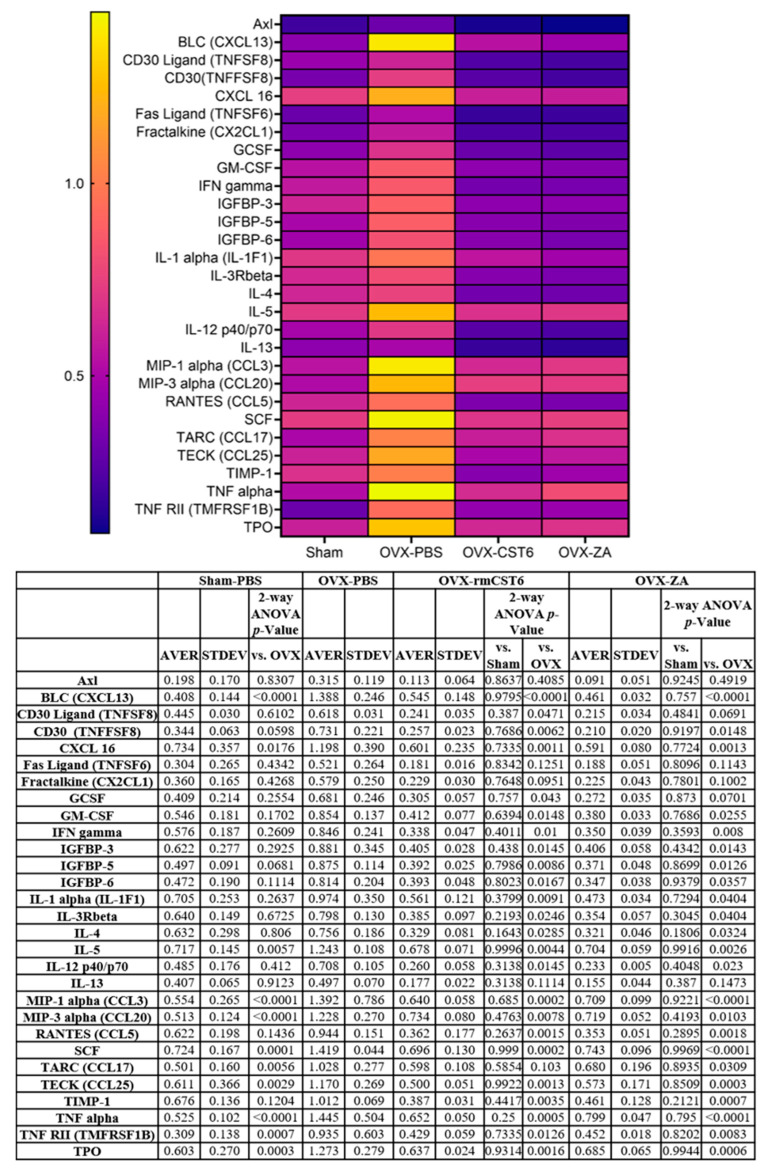
OVX-induced increase in inflammatory cytokine expression is brought back to sham levels by treatment with either rmCst6 or ZA. Inflammatory cytokine array for Protein samples from L3 to L5 mice vertebrae. The 25 targets with the largest initial change in membrane intensity from the sham surgery group to the OVX-PBS group are shown. Ovariectomy causes the upregulation of multiple inflammatory cytokines, which in turn can cause increased bone resorption. Treatment with either rmCst6 or ZA brought levels back to sham. For each group (sham, OVX-PBS, OVX-CST6, OVX-ZA), L3–L5 vertebrae protein lysate from each sample in the group was pooled. For pooled samples, triplicate analysis was performed.

**Figure 6 cancers-17-00833-f006:**
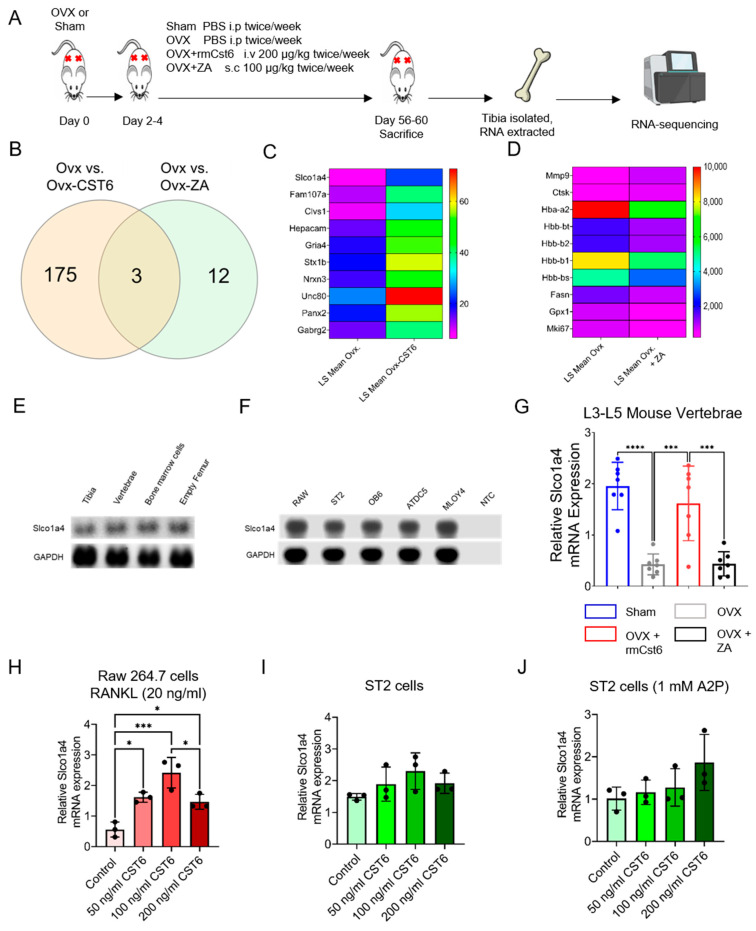
RNA-seq analysis of tibia from OVX-mouse model reveals potential different pathways for CST6 and ZA treatment. (**A**) Experimental workflow for RNA-seq experiment on tibia isolated from OVX mice. The red cross represents the site of s.c. injection. (**B**) One hundred seventy-five unique differentially expressed genes were found for OVX-CST6 treated tibia compared with the twelve differentially expressed genes for OVX-ZA-treated tibia. (**C**) The top 10 genes with largest fold increase in the tibia following rmCst6 treatment of OVX mice are listed in a heat map and sorted by LS Mean. Of interest, *Slco1a4*, an organic anion transporter that may be involved in bringing estrogen into the cell, had the largest fold increase (log2 fold = 3.60). (**D**) The top 10 genes with the greatest fold change (increase or decrease) following ZA treatment of OVX tibia are listed in a heat map sorted by LS Mean. (**E**,**F**) Basal tissue and cell level gene expression of *Slco1a4* determined by PCR. (**G**) RNA-seq results for tibia were confirmed using real-time PCR on RNA isolated from L3 to L5 vertebrae (*n* = 7). (**H**–**J**) Slco1a4 mRNA levels increase for both pre-osteoclastic Raw 264.7 cells treated with RANKL (20 ng/mL) as well as for pre-osteoblastic ST2 cells (±1 mM A2P) (*n* = 3). Only the increase for Raw 264.7 cells was considered significant. Data represented as mean ± SD analyzed by one-way ANOVA with Tukey’s multiple-comparisons. Denoted markings are considered significant * *p* < 0.05, *** *p* < 0.001, **** *p* < 0.0001.

**Figure 7 cancers-17-00833-f007:**
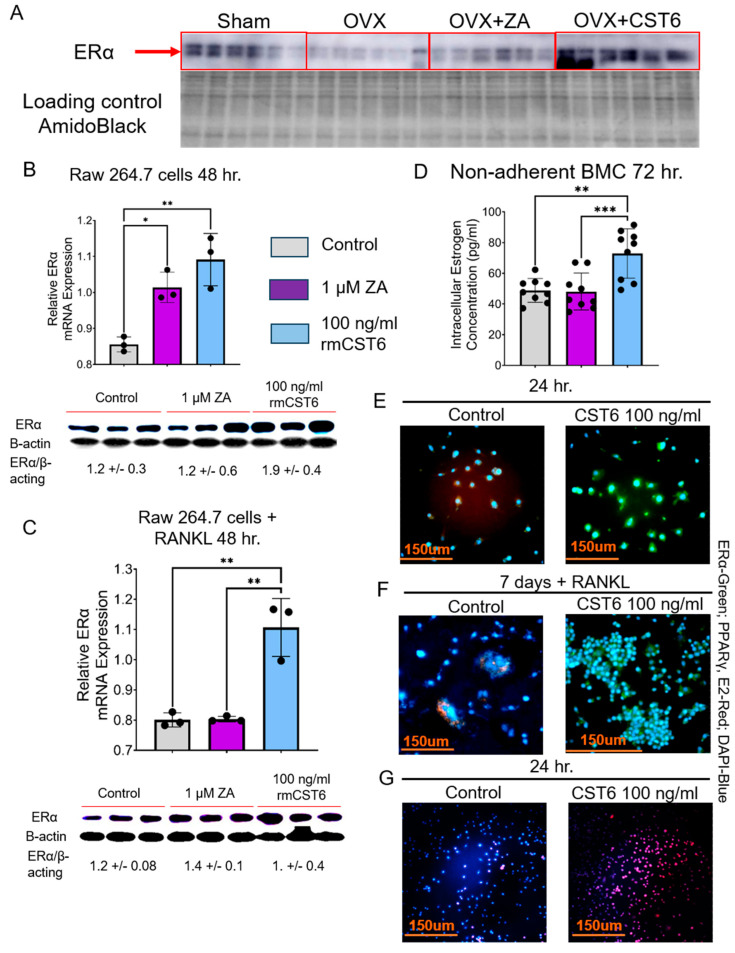
ERα expression in bone tissue and cells and intracellular total estrogen levels in osteoclast precursors are increased following treatment with rmCST6. (**A**) ERα protein levels are decreased in L3–L5 vertebrae of OVX mice, and expression is brought back to sham surgery levels only by rmCst6 treatment (*n* = 6). Amido black staining was used as a loading control. (**B**) Raw cells treated with either 1 µM ZA or 100 ng/mL of rmCST6 for 48 h had increased mRNA and protein levels of ERα when compared with control-treated samples (*n* = 3). Protein level significance was analyzed using one-tailed *t*-test (Table 4). (**C**) Raw cells treated with 1 µM ZA or 100 ng/mL of rmCst6 for 48 h (20 ng/mL RANKL) had increased mRNA and protein levels of ERα when compared with control-treated samples (*n* = 3). Protein level significance was analyzed using one-tailed *t*-test (Table 5). (**D**) Total intracellular estrogen levels (E1, E2, E3) for cultured osteoclast precursor cells isolated from mouse femur were increased for only 100 ng/mL rmCst6 treatment. Estrogen concentration was normalized to total protein lysate concentration. Graph is the result of triplicate experiments (*n* = 9). Three ELISA plates were scaled to each other by dividing estrogen concentration (pg/µg) for each sample by average estrogen concentration of control treatment for said plate, then multiplying by the average estrogen concentration for all samples across three plates. (**E**) Immunohistology staining of Raw 264.7 cells incubated for 24 h treated with PBS control solution or 100 ng/mL rmCST6. Cells treated with rmCst6 showed increased expression of ERα, whose transcription is known to be upregulated by estrogen, and decreased expression of PPARγ, whose transcription is known to be downregulated by estrogen. (**F**) Immunohistology staining of Raw 264.7 cells incubated for 7 days treated with PBS control solution or 100 ng/mL rmCst6 (20 ng/mL RANKL). Cells treated with rmCst6 showed increased expression of ERα and decreased expression of PPARγ. (**G**) Estrogen influx is increased in Raw 264.7 cells by 24 h rmCst6 treatment. Raw 264.7 cells were plated onto tissue culture-treated slides (Lab-Tek Chamber Slide System, 177437; Nalge Nunc International, Naperville, IL 60563, USA) at a concentration of 2.5 × 10^5^ cells/mL and treated with PBS (**left** panel) or 100 ng/mL rmCst6 (**right** panel) and 1 pm Estradiol glow (E2 Glow, Jena Bioscience, PR-958S) for 24 h. After 24 h, media was aspirated, and cells were fixed with cold 4% paraformaldehyde for 20 min and rinsed with PBS. Following this, cells were covered with DAPI-Fluoromount-G and observed using a Nikon Eclipse T/2 epifluorescent microscope. DAPI ex wavelength: 350 nm, em wavelength: 470 nm; E2 Glow ex wavelength: 467 nm, em wavelength: 618 nm. Data represented as mean ± SD analyzed by one-way ANOVA with Tukey’s multiple comparisons. Denoted markings are considered significant * *p* < 0.05, ** *p* < 0.01, *** *p* < 0.001.

**Table 1 cancers-17-00833-t001:** Lesion number and µCT parameters calculated for MM mouse models (*n* = 5).

	MM	MM + rmCST6	MM + ZA
Lesion Number	19.2 ± 4.2	12.6 ± 2.8	4.4 ± 0.9
BV/TV (%)	2.64 ± 0.52	3.57 ± 0.14	3.88 ± 0.53
Tb.N (1/mm)	0.56 ± 0.09	0.82 ± 0.02	0.85 ± 0.12
Tb.Th (mm)	0.047 ± 0.004	0.044 ± 0.002	0.045 ± 0.0008
Tb.Sp (mm)	0.45 ± 0.06	0.38 ± 0.02	0.38 ± 0.025
BMD (g/cm^3^)	0.072 ± 0.01	0.10 ± 0.008	0.12 ± 0.012

**Table 2 cancers-17-00833-t002:** µCT parameters calculated for OVX mouse models (*n* = *7*).

	Sham	OVX	OVX + rmCst6	OVX + ZA
BV/TV (%)	3.1 ± 0.6	2.2 ± 0.40	3.3 ± 0.43	3.84 ± 0.23
Tb.N (1/mm)	0.64 ± 0.08	0.50 ± 0.10	0.70 ± 0.078	0.86 ± 0.053
Tb.Th (mm)	0.049 ± 0.002	0.043 ± 0.002	0.043 ± 0.0008	0.04 ± 0.001
Tb.Sp (mm)	0.37 ± 0.03	0.39 ± 0.01	0.31 ± 0.01	0.37 ± 0.014
BMD (g/cm^3^)	0.087 ± 0.015	0.052 ± 0.0068	0.079 ± 0.008	0.10 ± 0.007

**Table 3 cancers-17-00833-t003:** Cytokine array data and statistical analysis (*n* = 3).

Protein	Sham Average	OVX-PBS Average	OVX-CST6 Average	OVX-ZA Average	*t*-Test Sham vs. OVX-PBS	*t*-Test OVX-CST6 vs. OVX-PBS	*t*-Test OVX-ZA vs. OVX-PBS
BLC (CXCL13)	0.41 ± 0.14	1.39 ± 0.25	0.54 ± 0.14	0.46 ± 0.03	0.004	0.004	0.001
MIP-1 alpha (CCL3)	0.55 ± 0.27	1.39 ± 0.78	0.64 ± 0.05	0.70 ± 0.09	0.07	0.08	0.10
MIP-3 alpha (CCL20)	0.51 ± 0.12	1.22 ± 0.27	0.73 ± 0.08	0.71 ± 0.05	0.01	0.13	0.13
SCF	0.72 ± 0.17	1.42 ± 0.04	0.69 ± 0.13	0.74 ± 0.096	0.001	0.000	0.000
TARC (CCL17)	0.50 ± 0.16	1.02 ± 0.27	0.59 ± 0.10	0.68 ± 0.19	0.03	0.03	0.07
TECK (CCL25)	0.61 ± 0.37	1.17 ± 0.26	0.50 ± 0.05	0.57 ± 0.17	0.05	0.007	0.01
TIMP-1	0.68 ± 0.14	1.01 ± 0.06	0.38 ± 0.03	0.46 ± 0.12	0.02	0.000	0.001
TNF alpha	0.53 ± 0.10	1.44 ± 0.50	0.65 ± 0.05	0.79 ± 0.04	0.01	0.027	0.04
TPO	0.60 ± 0.27	1.27 ± 0.27	0.63 ± 0.02	0.68 ± 0.06	0.02	0.008	0.01

**Table 4 cancers-17-00833-t004:** Western blot data and statistical analysis for Raw 264.7 cells (48 h incubation, Control, 1 µM ZA or 100 ng/mL rmCst6 treatment) (*n* = 3).

	Control Average	ZA Average	100 ng/mL CST6 Average	*t*-Test Control vs. ZA	*t*-Test Control vs. 100 ng/mL CST6	*t*-Test ZA vs. 100 ng/mL CST6
ERα band Intensity	13,735.0 ± 3336.7	15,366.5 ± 7562.1	24,189.5 ± 5269.1	0.37	0.021	0.086
β-actin band Intensity	11,835.3 ± 877.9	13,207.1 ± 837.6	13,846.5 ± 2350.0	0.06	0.11	0.34
ERα/β-actin	1.18 ± 0.32	1.17 ± 0.63	1.89 ± 0.39	0.48	0.036	0.084

**Table 5 cancers-17-00833-t005:** Western blot data and statistical analysis for Raw 264.7 cells (48 h incubation, 20 ng/mL RANKL) Control, 1 µM ZA or 100 ng/mL rmCst6 treatment) (*n* = 3).

	Control Average	ZA Average	100 ng/mL CST6 Average	*t*-Test Control vs. ZA	*t*-Test Control vs. 100 ng/mL CST6	*t*-Test ZA vs. 100 ng/mL CST6
ERα band Intensity	20,647.3 ± 2366.8	24,660.8 ± 1917.5	29,535.7 ± 3279.3	0.042	0.0094	0.045
β-actin band Intensity	16,643.0 ± 1369.6	17,405.5 ± 588.6	17,153.1 ± 1755.3	0.21	0.35	0.41
ERα/β-actin	1.23 ± 0.077	1.41 ± 0.096	1.74 ± 0.37	0.033	0.041	0.10

## Data Availability

The data that support the findings of this study are available in the Appendix A of this article.

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
