# Peer review of "Cystatin M/E Ameliorates Multiple Myeloma-Induced Hyper Osteolytic Bone Resorption"

_cancers, 2025, doi:10.3390/cancers17050833_

Round 1

Reviewer 1 Report

Comments and Suggestions for Authors

Osteolysis in multiple myeloma is an area of great clinical interest as this phenomenon may lead to disability. Currently zoledronic acid and denosumab are the 2 agents used to treat this condition. The authors of this interesting manuscript provide in-depth details into the possible involvement of cystatin, a protease inhibitor secreted by myeloma cells, especially in the subtype of low-bone-lesion.

Mouse experiments are done on ovariectomized and myeloma mice to really examine nearly every aspect of the cystatine-osteoclast axis. The findings are interesting but it is not very clear to me how this may impact the care of human myeloma. The observed interaction with estrogen metabolism may have some human relevance but still unclear for me. More explanation of the relevance of this really interesting experimental work for human myeloma is needed.

The authors indicate the zoledronic acid and denosumab-induced osteonecrosis may be prevented through this cystatin-based pathway and provide some experimental evidence that supports it. Unfortunately, I am not aware of any good model that can help to study the osteonecrosis process itself. Maybe some experimental evidence can be gained on this field, as well.

Author Response

We would like to express our gratitude to the reviewer for dedicating their valuable time to reviewing our manuscript and for providing such encouraging comments and suggestions. The reviewer raised concerns regarding the potential impact of our findings on the care of human myeloma, as well as the relevance of the interaction between CST6 and estrogen metabolism. These are precisely the areas we are currently exploring in our research, where we are modifying CST6 to potentially reduce the growth of multiple myeloma (MM) tumors and MM-induced osteolysis. Additionally, we are investigating how CST6 interacts with estrogen or estrogen receptor signaling in the context of osteoclastogenesis.

Regarding the concern about zoledronic acid-induced osteonecrosis and whether it may be prevented through a cystatin-based pathway, we did not claim that osteonecrosis could be prevented through this pathway. Rather, we simply presented our observations that micro-CT imaging revealed a thick band of calcified trabeculae with altered architecture near the growth plate in response to ZA treatment, which was not observed with CST6 treatment. We are careful not to over-interpret our data or discuss conclusions without additional supporting evidence.

Reviewer 2 Report

Comments and Suggestions for Authors

The paper "Cystatin M/E ameliorates multiple myeloma-induced hyper osteolytic bone resorption" investigates the potential of recombinant mouse Cystatin M/E (rmCst6) as a novel agent for treating bone resorption, particularly in the context of multiple myeloma (MM). The paper suggests that rmCst6 could be a potential therapeutic agent for MM osteolytic lesions and other bone resorption disorders.

Here are specific recommendations for improving the manuscript:

  • Results, line 114: "Intravenous (i.v) injection of purified rmCst6 113 protein (200 µg/kg, twice per week) and subcutaneous (s.c) injection of ZA (100 µg/kg, twice per week) improved bone compared to PBS injection, while also significantly de- 115 creasing the number of osteolytic lesions in MM-bearing mice (Figure 1B and 1C)." Specify which bone parameters were improved (e.g., bone volume, density) to provide a more precise understanding of the treatment effects.
  • Results, line 175: "after 6 weeks of ZA treatment, quantitative statistical histomorphometry showed a significant increase in TRAPase positive cells in the trabecular bone region, especially on the 176 surface of calcified cartilage that fills the tibia metaphysis (Figure 2D and 2E)." Explain the potential reasons and implications of this increase in TRAPase positive cells with ZA treatment. Is this an expected side effect, or does it indicate a specific mechanism of action?
  • Results, line 211: "When compared to other groups, only rmCst6 treated 211 mice showed a decrease in BM macrophage percentage and an increase in BM monocyte 212 percentage (Figure 3A and 3B)." Provide context or potential explanations for why rmCst6 treatment leads to these specific changes in bone marrow cell populations. What is the significance of these changes in the context of MM bone disease?
  • Results, line 331: "Of particular interest, for rmCst6 treatment, the solute carrier organic 331 anion transporter family member 1a4 (Slco1a4), an organic anion transporter, had the larg- 332 est fold increase (log2 fold = 3.60), while for ZA treatment, MMP9, which plays a role in 333 apoptotic pathways, had the greatest fold increase (log2 fold = 3.33)22,23." Since Slco1a4 and MMP9 are highlighted, briefly explain their known or hypothesized roles in bone metabolism or MM to provide a clearer rationale for their importance.
  • Discussion, line 475: "As M5 and M7 macrophages are thought to play a role in the 476 immune system, Cst6 may impact M5 and M7 macrophage levels through inhibition of 477 key cysteine proteases involved in immune system regulation35." Elaborate on the specific cysteine proteases that CST6 inhibits and how this inhibition could affect M5 and M7 macrophage levels. This would strengthen the proposed mechanism.
  • Discussion, line 514: "The mechanism by which rmCst6 promotes estrogen influx in macrophage precur- 514 sors is currently unknown. Based on OVX model RNA-seq data, expression of the organic 515 anion transport protein Slco1a4 is increased following treatment with rmCst6." Since the mechanism is unknown, clarify whether the increased expression of Slco1a4 is a correlation or if there is evidence suggesting a causal relationship. If it is speculative, use language that reflects this uncertainty.
  • Discussion, line 533: "However, a mechanism explaining how rmCst6 increases ERα expression is cur- 533 rently unknown. Previously it was proposed that ERβ may cause an upregulation of cys- 534 tatins in triple negative breast cancer47." Clarify the connection between ERβ, cystatins, and ERα expression to help the reader understand the proposed pathway.

These revisions should improve the manuscript by clarifying key findings, strengthening the rationale for the study's conclusions, and enhancing the overall readability and impact of the research.

Author Response

The paper "Cystatin M/E ameliorates multiple myeloma-induced hyper osteolytic bone resorption" investigates the potential of recombinant mouse Cystatin M/E (rmCst6) as a novel agent for treating bone resorption, particularly in the context of multiple myeloma (MM). The paper suggests that rmCst6 could be a potential therapeutic agent for MM osteolytic lesions and other bone resorption disorders.

Here are specific recommendations for improving the manuscript:

  • Results, line 114: "Intravenous (i.v) injection of purified rmCst6 113 protein (200 µg/kg, twice per week) and subcutaneous (s.c) injection of ZA (100 µg/kg, twice per week) improved bone compared to PBS injection, while also significantly de- 115 creasing the number of osteolytic lesions in MM-bearing mice (Figure 1B and 1C)." Specify which bone parameters were improved (e.g., bone volume, density) to provide a more precise understanding of the treatment effects.
  • Results, line 175: "after 6 weeks of ZA treatment, quantitative statistical histomorphometry showed a significant increase in TRAPase positive cells in the trabecular bone region, especially on the 176 surface of calcified cartilage that fills the tibia metaphysis (Figure 2D and 2E)." Explain the potential reasons and implications of this increase in TRAPase positive cells with ZA treatment. Is this an expected side effect, or does it indicate a specific mechanism of action?
  • Results, line 211: "When compared to other groups, only rmCst6 treated 211 mice showed a decrease in BM macrophage percentage and an increase in BM monocyte 212 percentage (Figure 3A and 3B)." Provide context or potential explanations for why rmCst6 treatment leads to these specific changes in bone marrow cell populations. What is the significance of these changes in the context of MM bone disease?
  • Results, line 331: "Of particular interest, for rmCst6 treatment, the solute carrier organic 331 anion transporter family member 1a4 (Slco1a4), an organic anion transporter, had the larg- 332 est fold increase (log2 fold = 3.60), while for ZA treatment, MMP9, which plays a role in 333 apoptotic pathways, had the greatest fold increase (log2 fold = 3.33)22,23." Since Slco1a4 and MMP9 are highlighted, briefly explain their known or hypothesized roles in bone metabolism or MM to provide a clearer rationale for their importance.
  • Discussion, line 475: "As M5 and M7 macrophages are thought to play a role in the 476 immune system, Cst6 may impact M5 and M7 macrophage levels through inhibition of 477 key cysteine proteases involved in immune system regulation35." Elaborate on the specific cysteine proteases that CST6 inhibits and how this inhibition could affect M5 and M7 macrophage levels. This would strengthen the proposed mechanism.
  • Discussion, line 514: "The mechanism by which rmCst6 promotes estrogen influx in macrophage precur- 514 sors is currently unknown. Based on OVX model RNA-seq data, expression of the organic 515 anion transport protein Slco1a4 is increased following treatment with rmCst6." Since the mechanism is unknown, clarify whether the increased expression of Slco1a4 is a correlation or if there is evidence suggesting a causal relationship. If it is speculative, use language that reflects this uncertainty.
  • Discussion, line 533: "However, a mechanism explaining how rmCst6 increases ERα expression is cur- 533 rently unknown. Previously it was proposed that ERβ may cause an upregulation of cys- 534 tatins in triple negative breast cancer47." Clarify the connection between ERβ, cystatins, and ERα expression to help the reader understand the proposed pathway.

These revisions should improve the manuscript by clarifying key findings, strengthening the rationale for the study's conclusions, and enhancing the overall readability and impact of the research.

Author’s response: We sincerely thank the reviewer for dedicating their valuable time to reviewing our manuscript and providing such detailed and specific recommendations. We have carefully considered all of the reviewer’s suggestions and have incorporated them into the revised manuscript, as outlined with corresponding line numbers.

Round 2

Reviewer 2 Report

Comments and Suggestions for Authors

The Authors adequately addressed my concerns. 

Comments on the Quality of English Language

English is fine.